# DIFFERENTIABLE HYPER-PARAMETER OPTIMIZATION

## ABSTRACT

Hyper-parameters are widely present in machine learning. Concretely, large amount of hyper-parameters exist in network layers, such as kernel size, channel size and the hidden layer size, which directly affect performance of the model. Thus, hyper-parameter optimization is crucial for machine learning. Current hyper-parameter optimization always requires multiple training sessions, resulting in a large time consuming. To solve this problem, we propose a method to fine-tune neural network's hyper-parameters efficiently in this paper, where optimization completes in only one training session. We apply our method for the optimization of various neural network layers' hyper-parameters and compare it with multiple benchmark hyper-parameter optimization models. Experimental results show that our method is commonly 10 times faster than traditional and mainstream methods such as random search, Bayesian optimization and many other state-of-art models. It also achieves higher quality hyper-parameters with better accuracy and stronger stability.

## 1 INTRODUCTION

HPO(hyper-parameter optimization) is one of the most critical parts in auto-ML (Thornton et al., 2012; Domhan et al., 2015; Kotthoff et al., 2017). Simple and low-dimensional hyper-parameters can be adjusted manually. It is also practical to use grid search or combine grid search with manual adjustment to deal with this simple problem (Montavon et al., 2012; gri, 2007; Hinton, 2010).

With the number of hyper-parameters continues increasing, manual tuning and grid search get ineffective. For a slightly large neural network model, we can use BO(bayesian optimization) (Snoek et al., 2012) or ZOOpt(zeroth-order optimization) (Liu et al., 2018) to optimize hyper-parameters. BO uses Gaussian process regression to fit mean and variance of objective function. However, a large number of matrix operations are needed in the fitting process, and multiple training sessions are needed to evaluate the predicted hyper-parameters. Time consuming increases along with each training session, and the total time consuming is positively correlated with the number of training sessions. Therefore, BO is still not suitable for large amount of hyper-parameters and so is ZOOpt.

In addition to BO and ZOOpt, many evolutionary algorithms (Young et al., 2015) have also been applied. For example, genetic algorithm (Goldberg) is often used for HPO. Evolutionary algorithms can often avoid the problem of local optimal. However, a fatal flaw is the low time efficiency.

When dealing with large-scale systems, random search (ran, 2012) is a practical and more efficient method to solve HPO. Based on experiences, random search may perform better than BO in some cases with better time efficiency and accuracy. In addition, currently very effective HPO methods are probably HB(Hyperband (Li et al., 2016) and DEHB (Awad et al., 2021). They accelerate the convergence and make it twice faster than random search.

Totally, all current models have a common defect which is a requirement for multiple training sessions of the fine-tuned model. While each training session is always expensive in time consuming. To get rid of this limit, we propose DHPO(differentiable hyper-parameter optimization), which accomplish HPO within one training session. Concretely, we change the form of hyper-parameters and include them in the calculation of forward propagation to achieve differentiable hyper-parameters. In this way, hyper-parameters are differentiable with the goal of minimize the loss function. When the session is over, hyper-parameters complete optimization together with network's parameters. Therefore, only one training session is needed in DHPO.

The contributions of this paper are summarized as follows.

- In this paper, we propose DHPO, which is the first model to solve hyper-parameter optimization in only one training session.
- The proposed approach is universal. It is an idea for HPO which can be applied to various kinds of hyper-parameters in neural network.
- We conduct extensive experiments on various kinds of neural network layers compared with multiple benchmark HPO models to evaluate the extreme efficiency and high performance of DHPO.

In the remaining of this paper, Section 3 describes the proposed method. Experiments are conducted and analyzed in Section 4. We overview related work in Section 5. Section 6 draws the conclusions and future work.

## 2 BACKGROUND

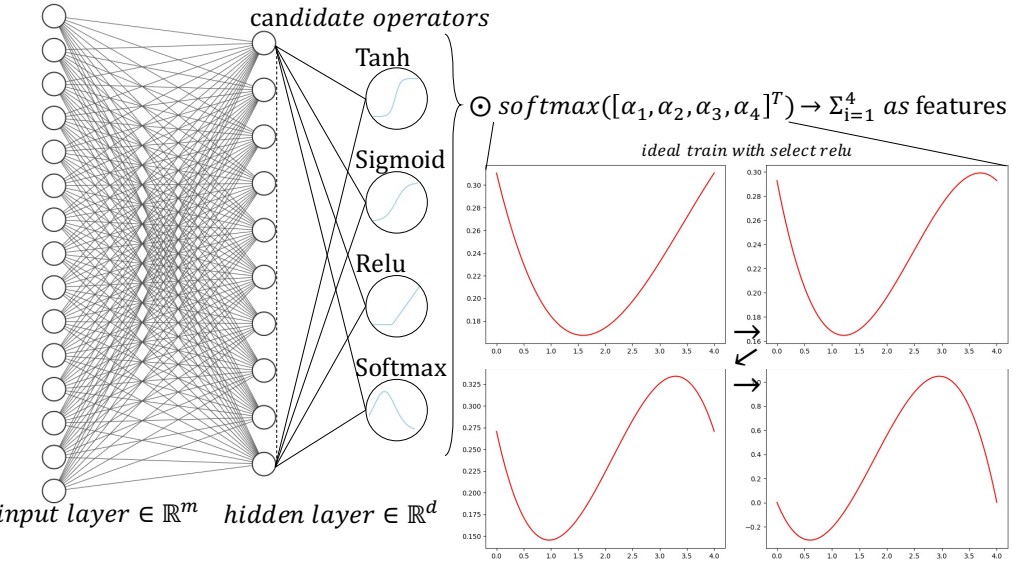

Figure 1: Select an activation with Darts

In this section, we introduce related techniques briefly.

Darts (Liu et al., 2019) is used to solve automatic network architecture search, which is described to determine the best operation from multiple candidate operators locally and globally. Take Figure 1 as example. It is selecting an activation for the fully connected layer from $\{Tanh, Sigmoid, Relu, Softmax\}$ and supposing $Relu$ is the best choice. Grid search costs four training sessions to evaluate each activation. While Darts can select the hyper-parameters in a single training session. Firstly, Darts calculates features $\mathcal{X}$ transformed by the fully connected layer and then uses each activation to process $\mathcal{X}$ once to get four non-linear features, denoted as $\boldsymbol{o} = [o_1, o_2, o_3, o_4]$. Further, $softmax(\boldsymbol{\alpha})$ assigns a weight to each $o_i$ and aggregates weighted $o_i$ by summing them, where $\alpha_i$ is a trainable parameter. An ideal train always ends up with $\max softmax(\boldsymbol{\alpha}) \to 1$ as shown in Figure 1. Darts in Figure 1 finally determines $Relu$ as the optimal activation with $softmax(\boldsymbol{alpha})_3$ very close to 1. Accordingly, $softmax(\boldsymbol{\alpha})_i \approx 0, i \neq 3$ meaning a shield to the information contained in $o_1, o_2, o_4$, which is equivalent to using $Relu$ only.

## 3 METHOD

As introduced in the background, Darts still has fatal defects. On the one hand, $\max softmax(\boldsymbol{\alpha})$ doesn't always converge to a level very close to 1. The cases where multiple candidates occupy

similar weights always exist especially for a large amount of candidates, which means Darts fails to distinguish different candidates and hit the optimal. On the other hand, Darts is only suitable for hyper-parameter with independent candidates. For example, the channel size is beyond Darts's capability. If we'd like to apply Darts to solve channel size, we should set a candidate for each possible value of channel size. Then there will be $O(n^2)$ channels in total, which is space expensive.

Motivated by this, we propose DHPO, aiming at solving all these problems existing in Darts and current HPO models. In the following, we first take the optimization of convolution layer's channel size as example to draw the core of DHPO in Section 3.1. Then, we apply DHPO to more hyper-parameters in Section 3.2.

### 3.1 DIFFERENTIABLE HYPER-PARAMETER OPTIMIZATION

The target of DHPO is to make hyper-parameter $\theta$ differentiable, and we achieve it through constructing trainable parameter $\alpha$ to substitute $\theta$ in the training. Obviously, the core of the idea is to point out the limitations for $\alpha$ and give out a universal method to construct and apply $\alpha$.

In this section, we first declare the sufficient conditions for $\alpha$ to substitute $\theta$ based on Theorem 1 in Section 3.1.1. Then we explain how to construct $\alpha$ and use it to control the structure of neural network, namely that is the forward propagation under $\alpha$ in Section 3.1.2. In the end, we use Theorem 1 to prove that $\alpha$ is able to substitute $\theta$ as Theorem 2 in Section 3.1.3.

#### 3.1.1 LIMITATIONS FOR $\alpha$

Our target is to construct trainable parameter $\alpha$ to substitute $\theta$ in the training. We will point out the sufficient conditions for $\alpha$ in the following.

Firstly, we define found sufficient conditions as a new relation between $\alpha$ and $\theta$, which is defined as *expressible* as Definition 3.1. Then, we prove that if $\alpha$ is expressible for $\theta$, then we can replace $\theta$ with $\alpha$ in neural network as Theorem 1. The definition and theorem are as follows.

**Definition 3.1.** ***Expressible***: *For a neural network $\mathcal{F} : \mathbb{R}^d \mapsto \mathbb{R}$ with $W$ and $\Theta$ as parameters and hyper-parameters respectively, hyper-parameter $\theta_2 \in \Omega_2$ is expressible for hyper-parameter $\theta_1 \in \Omega_1$ if and only if there is a surjective $h : \Omega_2 \mapsto \Omega_1(\theta_1 = h(\theta_2))$ which makes network structure under $\mathcal{F}(W|\theta_1)$ and that under $\mathcal{F}(W|\theta_2)$ are the same.*

**Theorem 1.** *If $\theta_2$ is **expressible** for $\theta_1$, then we can replace $\theta_1$ with $\theta_2$ in neural network.*

*Proof.* If $\theta_2$ is expressible for $\theta_1$, then there is a surjective from $\theta_2$ to $\theta_1$. Therefore, $\forall \theta_1 \in \Omega_1, \exists \theta_2 \in \Omega_2, h(\theta_2) = \theta_1$. Meanwhile, $\mathcal{F}(W|\theta_2)$ and $\mathcal{F}(W|\theta_1)$ share the same structures according to Definition 3.1. So we can replace $\theta_1$ with $\theta_2$ in a neural network. □

According to Theorem 1 and Definition 3.1, we can replace $\theta$ with $\alpha$ in neural network as long as the constructed $\alpha$ satisfies the following two conditions. One is that there is a surjective $\theta = h(\alpha)$. The other is that the network structure controlled by $\alpha$ is the same to that of $\theta$ under $\theta = h(\alpha)$.

#### 3.1.2 FORWARD PROPAGATION UNDER $\alpha$

In this section, we first propose the method to construct expressible $\alpha$ as described above, and then explain how $\alpha$ controls the structure of neural network, i.e., applying $\alpha$ in the forward propagation.

$$\phi(X|\boldsymbol{\alpha}) = \sum_{i=1}^{sup(\theta)} \epsilon_i * (X \star \kappa_i)$$

$$\boldsymbol{\epsilon} = \sigma((softmax(\boldsymbol{\alpha})) \cdot A - \sigma(\beta)) * a) \tag{1}$$

We take convolution layer's channel size as example to construct trainable $\boldsymbol{\alpha}$, which should be expressible for $\theta_{channel\_size}$(abbreviated as $\theta$). We construct $\boldsymbol{\alpha}$ as $\boldsymbol{\alpha} = [\alpha_1, \cdots, \alpha_{sup(\theta)}]$ and $\boldsymbol{\alpha} \in \mathbb{R}^{sup(\theta)}, sup(\theta) = \max \theta$. Formula 1 describes the forward propagation in this convolution layer under $\boldsymbol{\alpha}$. $X \in \mathbb{R}^{W \times H}$ is the input of convolution layer which is a single-channel image. $\kappa_i$ represents the convolution kernel of $i$-th channel, and we prepare $sup(\theta)$ candidate kernels in the initialization of neural network. ($\star$) represents convolution operation and ($\cdot$) represents matrix

multiplication. $A = \begin{bmatrix} 1 & \cdots & 1 \\ & \ddots & \vdots \\ \mathbf{0} & & 1 \end{bmatrix}^{\mathrm{T}}$ is a lower triangular matrix. $\gamma \in \mathbb{R}$ is a trainable parameter and $a \in \mathbb{R}$ is a large constant.

Obviously, the core of formula 1 is the calculation of $\boldsymbol{\epsilon}$. We visualize the calculation in Figure 2. Two cases are given in this figure. We take the left as example. $\boldsymbol{\alpha}$ is first transformed by softmax activation and $0 < \text{softmax}(\boldsymbol{\alpha})_i < 1$. Then we get $\mathcal{A}$ by multiplying matrix $softmax(\boldsymbol{\alpha})$ and matrix $A$, $\mathcal{A}_1 = 1$ and $\mathcal{A}_i > \mathcal{A}_j$ if $i > j$. Next, we subtract $\sigma(\gamma)$ from $\mathcal{A}_i$ and multiply it by a large constant $a$, $a > 1$. In this time, $-a < (\mathcal{A}_i - \sigma(\gamma)) * a < a$, and it maintains the same monotonicity as $\mathcal{A}$. In the last step, we use $\sigma$ again to map $(\mathcal{A}_i - \sigma(\gamma)) * a$ to a value close to $1$ or $0$. Finally, $\epsilon_i = \sigma((\mathcal{A}_i - \sigma(\gamma)) * a)$. In this example, $\epsilon_1, \epsilon_2, \epsilon_3$ are very close to $1$, and the others are very close to $0$. In this occasion, we think that the channel size is 3.

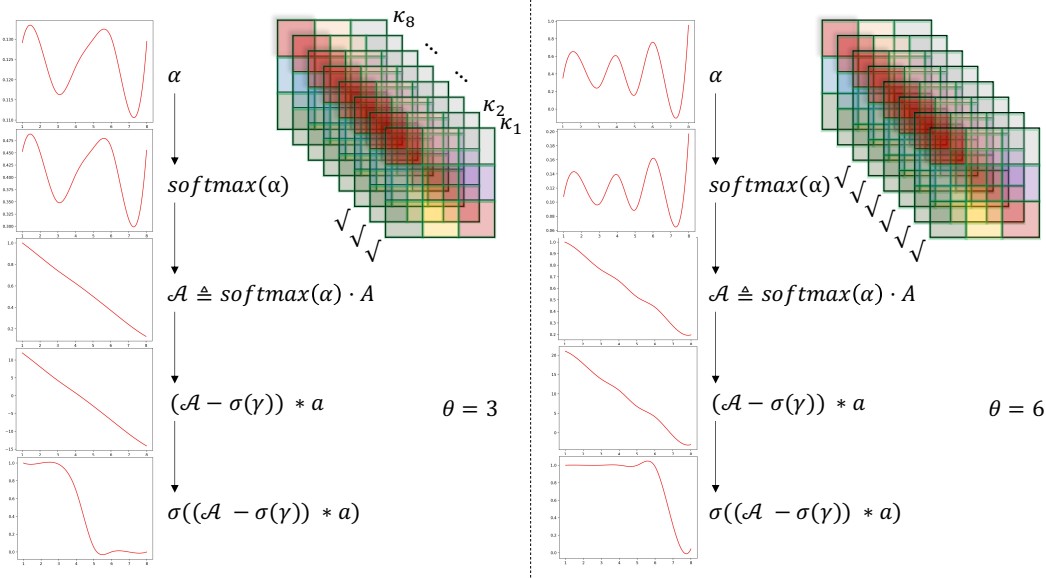

Figure 2: Forward propagation under $\boldsymbol{\alpha}$ in DHPO.

Based on the above construction, $\boldsymbol{\alpha}$ has the same effect as $\theta_{channel\_size}$. Thus, it is reasonable to substitute $\theta_{channel\_size}$ with $\boldsymbol{\alpha}$. We will prove this in the following Section 3.1.3.

### 3.1.3 EXPRESSIBLE FOR $\theta$

The goal is to prove that we can substitute $\theta_{channel\_size}$ with $\boldsymbol{\alpha}$. According to Theorem 1, we just need to prove that $\boldsymbol{\alpha}$ is expressible for $\theta_{channel\_size}$. Before proof, we first list three key properties of $\boldsymbol{\epsilon}$ in the following which are necessary to the proof.

$\boldsymbol{\epsilon}$ is constructed from $\boldsymbol{\alpha}$ and has three key characteristics. ① $\epsilon_1 \approx 1$. ② $\epsilon_1 > \cdots > \epsilon_{sup(\theta)}$. ③ There is an index $t$ dividing $\boldsymbol{\epsilon}$ into two sets $\varepsilon_{big} = \{\epsilon_i | 1 \le i \le t\}$ and $\varepsilon_{small} = \{\epsilon_i | t + 1 \le i \le sup(\theta)\}$. The items in $\varepsilon_{big}$ are all close to $1$ and items in $\varepsilon_{small}$ are all close to $0$. The first property ensures that at least one channel is selected. The second property ensures the selected channels are always the first few rather than random several in candidate channels. The third property makes unselected channels blocked away, especially when $a \mapsto \infty$. Based on these features, we have Theorem 2.

**Theorem 2.** *When $a \mapsto \infty$, $\boldsymbol{\alpha}$ is expressible for $\theta$ under $h(\boldsymbol{\alpha}) = rounded \sum \boldsymbol{\epsilon}$.*

*Proof.* If we would like to prove $\boldsymbol{\alpha}$ is expressible for $\theta$, we just need to prove $h(\boldsymbol{\alpha})$ is surjective and network structure controlled by $\boldsymbol{\alpha}$ is the same to that of $\theta$ under $\theta = h(\alpha)$ according to the discussion in Section 3.1.1.
Obviously, $\theta = h(\boldsymbol{\alpha})$ is a surjective. We then prove convolution layers under $\theta$ and $\boldsymbol{\alpha}$ have same

structure. Suppose that the network structure controlled by $\theta$ is a convolution layer composed of the first $\theta$ channels from candidates. As for the structure controlled by $\boldsymbol{\alpha}$, we can also see it as a convolution layer composed of the first $\theta$ channels according to the third characteristic of $\boldsymbol{\epsilon}$. Because noise from unselected channels will fade out with $a \to \infty$. At this time, $\mathcal{F}(X|\boldsymbol{\alpha}) \Leftrightarrow \mathcal{F}(X|\theta)$. $\qquad \square$

Based on above discussions, we finally achieve the differentiable channel size. Since $\boldsymbol{\alpha}$ is trainable and expressible for $\theta_{channel\_size}$. Then we can replace $\theta_{channel\_size}$ with $\boldsymbol{\alpha}$ in this convolution layer.

When the training session is over, we take rounded $\sum \boldsymbol{\epsilon}$ as the optimized $\theta_{channel\_size}$.

## 3.2 Multiple hyper-parameters optimization

Based on the idea discussed in Section 3.1, we can apply DHPO to a large amount of hyper-parameters rather than just channel size.

In this section, we select two widely used and typical hyper-parameters to explain how to deal with them in DHPO, and other hyper-parameters can be processed in a similar way. These typical hyper-parameters are convolution kernel size and hidden layer size.

### 3.2.1 KERNEL SIZE

In this section, we introduce method of making kernel size differentiable. The challenge here is that we should deal with multi-dimensional hyper-parameter. Therefore, we should make DHPO suitable for multi-dimensional hyper-parameter rather than just one dimensional hyper-parameter like channel size. We achieve this by extending $\boldsymbol{\epsilon}$ to multiple dimensions and the detail is as follows.

Firstly, we construct $\boldsymbol{\epsilon}$ from $\boldsymbol{\alpha}$ as described in Section 3.1 and a candidate kernel $\kappa$. For the one-dimensional kernel, $\kappa \in \mathbb{R}^{sup(\theta_{kernel\_size})}$ ($\theta_{kernel\_size}$ abbreviated as $\theta$). And we use $\kappa \odot \boldsymbol{\epsilon}$ as the kernel in forward propagation. For the 2-dimensional kernel $\kappa \in \mathbb{R}^{sup(\theta) \times sup(\theta)}$, we use $\kappa \odot M$ as

selected kernel, $M = \begin{bmatrix} \epsilon_1 & \epsilon_2 & \cdots & \epsilon_{sup(\theta)} \\ \epsilon_2 & \epsilon_2 & \cdots & \epsilon_{sup(\theta)} \\ \vdots & \vdots & \ddots & \vdots \\ \epsilon_{sup(\theta)} & \epsilon_{sup(\theta)} & \cdots & \epsilon_{sup(\theta)} \end{bmatrix}$. In this way, the features from unselected items $\epsilon_i$ are blocked away since $\epsilon_i \approx 0$.

Then $\boldsymbol{\alpha}$ is expressible for $\theta$ under $h(\boldsymbol{\alpha}) = $ rounded $\sum \boldsymbol{\epsilon}$. Obviously, $h(\boldsymbol{\alpha})$ is a surjective. When the large constant $a \to \infty$. In this time, kernel size under $\theta$ and that under $\boldsymbol{\alpha}$ are the same under $\theta = h(\boldsymbol{\alpha})$. So we can substitute $\theta_{kernel\_size}$ with $\boldsymbol{\alpha}$ in neural network. When the training session is over, we take rounded $\sum \boldsymbol{\epsilon}$ as the optimized $\theta_{kernel\_size}$.

Based on convolution kernel size, we can design differentiable hyper-parameter for pooling kernel size, convolution stride and etc. in a similar way.

### 3.2.2 HIDDEN LAYER SIZE

In this section, we apply DHPO to one of the most common and important hyper-parameters, hidden layer size. Commonly, the hidden layer size has thousands of candidates. Therefore, it is difficult for sample-based models such as BO and random search. In contrast, DHPO can solve it easily both theoretically and practically since hidden layer size is differentiable in DSA.

Again, we first construct $\boldsymbol{\epsilon}$ from $\boldsymbol{\alpha}$ as described above and use $\boldsymbol{\epsilon}$ to control the hidden layer size. We use $(X \cdot W + \boldsymbol{b}) \odot \boldsymbol{\epsilon}$ as the output of the fully connected layer. $(X \cdot W + \boldsymbol{b})$ is the original output and its dimension is $sup(\theta_{hidden\_size})$. In this way, unselected last few features will be blocked away, which is equivalent to use a smaller hidden layer size. When the training session is over, we take rounded $\sum \boldsymbol{\epsilon}$ as optimized $\theta_{hidden\_size}$.

## 4 EXPERIMENT

To verify the effectiveness of the proposed approach, we conduct HPO on two different neural networks with multiple datasets, and compare it with various baselines. Section 4.1 briefly introduces

the neural network to be tuned, Section 4.2 gives the basic settings, Section 4.3 displays the experimental results, and Section 4.4 conducts a case study to show the DHPO tuning process.

## 4.1 MODEL TO BE TUNED

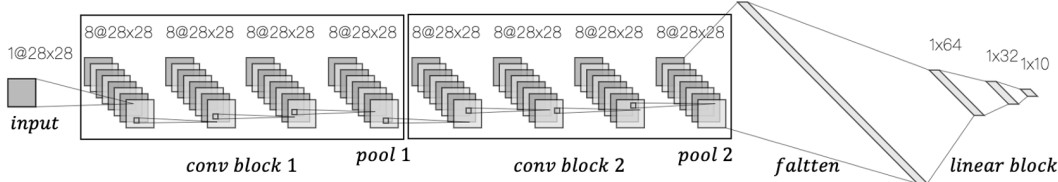

Figure 3: Deep convolution neural network to be tuned

In the experiment, we apply DHPO to two classical models, i.e. deep convolutional neural networks and MLP(multi-layer perceptron), which include various kinds of hyper-parameters. DNN(deep convolution neural network) is a lightweight convolution network designed with reference to Kunihiko & Fukushima (1980); Lecun & Bottou (1998); Behnke & Sven (2003). Even though DNN is not the best model to process images, it is a good network which can distinguish the performance of various HPO approaches since it has a large number of hyper-parameters. DNN's network structure is as shown in Figure 3, which is a lightweight implementation. And MLP is composed of 4 fully connected layers. DNN is used to evaluate HPO for convolution layer and pooling layer. MLP is used to evaluate HPO for fully connected layer.

## 4.2 EXPERIMENTAL SETTINGS

In this section, we introduce the experimental settings.

### 4.2.1 DATASET

MNIST (LeCun & Cortes, 2010) and SVHN (svh, 2011) are well known lightweight validation image datasets, and they are suitable for DNN. Therefore, we use MNIST and SVHN as the benchmark datasets on DNN. On MLP, we choose iris, wine, car and agaricus-lepiota[1]. Basic message of the six datasets is shown in Appendix A.

### 4.2.2 BASELINE

The mainstream algorithms used for HPO contains random search, BO, evolutionary algorithm and so on. We finally choose the following HPO algorithm as baselines:

- Random search (ran, 2012) is the most universal HPO algorithm, which can randomly select points in the entire search space. So it is able to break the limit of local optimal solution.
- Zoopt (Liu et al., 2018) does not rely on the gradient of the objective function, but learns from samples of the search space instead. It is suitable for HPO tasks that are not differentiable, with many local minimal or even unknown but only testable.
- Bayes optimization (Snoek et al., 2012) is based on Gaussian process regression to fit the distribution between samples and objective. It has a practical effect and wide range of applications.
- Genetic algorithm (Goldberg) belongs to evolutionary algorithm, relying on the mutation and genetic of maintained population. It is often used in HPO and NAS(network architecture search).
- Particle swarm algorithm (Kennedy & Eberhart, 1995) also belongs to evolutionary algorithm, which finds the optimal solution through collaboration and information sharing between individuals in the group. Simple implementation, few hyperparameters, and wide applications.
- HyperBand (Li et al., 2016) regards the process of finding the optimal hyper-parameters as a non-random exploration on the infinite arm bandit under the condition of limited resources.
- DEHB (Awad et al., 2021) is developed based on differential evolution and HB, which is furnished with better time performance and convergence effect.

---

[1]https://archive.ics.uci.edu/ml/datasets

### 4.2.3 METRICS

- Time consuming reflects the time interval from when data set and model are loaded on the GPU to when the algorithm stops. If data set is loaded multiple times, the loading time is also included.
- Accuracy reflects the performance of neural network generated by HPO models.
- Mean of top-K accuracy. HPO model always gives multiple sets of hyper-parameters and corresponding accuracy. We calculate the mean accuracy of top-K models.
- Loss sequence reflects the state of objective and convergence effect in a training session.

### 4.2.4 PARAMETER SETTINGS

**Tuned hyper-parameters** We fine-tune convolution layer's channel size, convolution layer's kernel size, pooling layer's pooling type and pooling layer's kernel size in DNN model, totally 16 hyper-parameters. Among them, $channel\ size \in [1, 8], kernel\ size \in [2, 5], pooling\ type \in [MaxPool, AvgPool]^2$. Additionally, 3 hidden layer size are optimized in MLP.

**DNN** We uses the Adam optimizer with learning rate = 0.001 and uses cross-entropy loss function. $batch\ size = 500$ and each training session contains 30 epochs. **MLP** All are the same to DNN except for 150 epochs in training session.

**BO** BO model is initialized with 5 samples, and continues iterating for 25 rounds. **ZOOpt** Iteration budgets is 30, and other hyper-parameters use the built-in configurations.[3] **Random search** The algorithm iterates 30 times. **Evolutionary algorithm** The population size and iterations are both 10, and mutation probability is 0.001. **HB** Iterate 81 times to keep fair with other models, because of its different iteration principles. **DEHB** We set min_budget and max_budget as 2 and 50, respectively.

Considering the influence of parameter's initialization, a three-times repeated independent experiment is conducted and our experiments are conducted on one GTX 3060Ti GPU.

### 4.3 EXPERIMENTAL RESULTS

Table 1: Time consuming and accuracy on MNIST and SVHN. Column 3-9 corresponds to various statistical accuracy(%). ⋆ denotes that we repeat DHPO 10 times, and † represents a once DHPO with three repeated independent experiments. BLANK corresponds to the blank group without any hyper-parameter optimization by setting all the channel size to 8 and set all the kernel size to 3.

| MODEL | TIME(s) | TOP1 | ALL MEAN | ALL STD | TOP5 MEAN | TOP5 STD | TOP10 MEAN | TOP10 STD |
|---|---|---|---|---|---|---|---|---|
| | | | | **MNIST** | | | | |
| BO | 6,298 | 98.66 | 97.32 | 1.14 | 98.51 | 0.11 | 98.33 | 0.2 |
| ZOOpt | 6,320 | 98.56 | 98.05 | 0.6 | 98.47 | **0.05** | 98.41 | **0.08** |
| RAND | 6,308 | 98.45 | 95.81 | 2.58 | 98.1 | 0.19 | 97.78 | 0.39 |
| HB | 13,918 | 96.17 | 82.82 | 13.04 | 98.44 | **0.05** | 98.37 | **0.08** |
| DEHB | 6,073 | 98.37 | 96.7 | 1.58 | 98.19 | 0.11 | 97.95 | 0.28 |
| DHPO⋆ | **1,708** | **99.01** | **98.57** | **0.19** | **98.7** | **0.05** | **98.57** | 0.19 |
| DHPO† | 512 | 98.71 | | | | | | |
| BLANK | | 98.27 | | | | | | |
| | | | | **SVHN** | | | | |
| BO | 11,043 | 87.29 | 83.3 | 3.95 | **86.81** | 0.27 | **86.49** | **0.38** |
| ZOOpt | 10,215 | 85.53 | 82.73 | 2.91 | 85.43 | **0.08** | 85.09 | 0.4 |
| RAND | 10,193 | 86.18 | 79.81 | 3.81 | 85.37 | 0.87 | 84.12 | 1.45 |
| DEHB | 10,194 | 85.61 | 81.18 | 4.51 | 85.15 | 0.28 | 84.45 | 0.73 |
| DHPO⋆ | **2,811** | **87.44** | **85.22** | **1.83** | 86.78 | 0.53 | 85.22 | 1.83 |
| DHPO† | 1,070 | 85.05 | | | | | | |
| BLANK | | 84.95 | | | | | | |

As shown in Table 1, DHPO can always complete tasks faster than any other HPO model several times with ensuring good performance. Taking MNIST as example, DHPO⋆ wins the game by taking only 28% of DEHB's time consuming. Additionally, DHPO† can also ensure the advantage in accuracy with fewer repeats, which means DHPO is stable. The stability can also be reflected from top-K accuracy's standard deviation, which is only 0.19 for all samples under DHPO.

---

[2]In DHPO, we invoke Darts to solve pooling type and we have proved it's effective here in Appendix B

[3]https://github.com/polixir/ZOOpt

Table 2: Time consuming and accuracy on small feature datasets.

| | IRIS | | WINE | | CAR | | AGARICUS | |
|---|---|---|---|---|---|---|---|---|
| **MODEL** | **TIME**(s) | **TOP1** | **TIME**(s) | **TOP1** | **TIME**(s) | **TOP1** | **TIME**(s) | **TOP1** |
| BO | 45 | 100 | 46 | 99.07 | 52 | 93.26 | 69 | 100 |
| ZOOpt | 44 | 100 | 45 | 100 | 47 | 92.87 | 71 | 100 |
| RAND | 38 | 100 | 45 | 100 | 48 | 93.16 | 64 | 100 |
| GA | 148 | 98.89 | 151 | 96.3 | 171 | 93.55 | 274 | 100 |
| PSO | 142 | 100 | 145 | 99.07 | 171 | 93.35 | 167 | 100 |
| HB | 24 | 100 | 24 | 87.04 | 26 | 69.36 | 40 | 99.94 |
| DEHB | 48 | 100 | 45 | 100 | 49 | 93.26 | 59 | 100 |
| DHPO⋆ | 9 | 100 | 9 | 100 | 10 | 92.49 | 16 | 100 |
| DHPO† | 3 | 100 | 3 | 96.3 | 3 | 91.43 | 5 | 100 |

On feature datasets shown in Table 2, DHPO also has absolute advantage in efficiency. While in some cases its accuracy is not the best compared to baselines. This is because DHPO trains the network while optimizing hyper-parameters resulting in a need for more epochs than other models. In this experiment, we set all epochs to be consistent in order to ensure the fairness of the experiment.

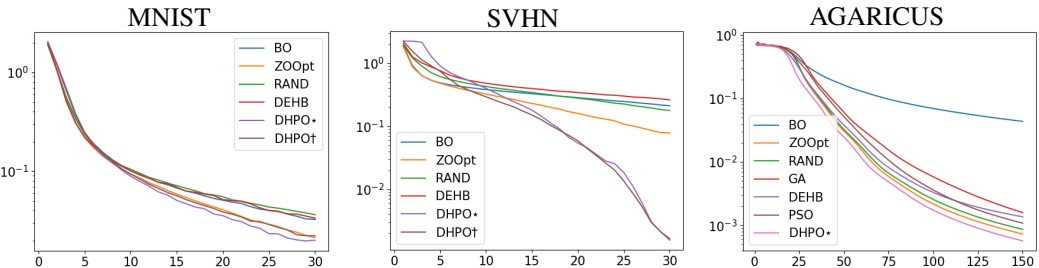

Figure 4: Loss sequence in the training session

As shown in Figure 4, DHPO shows a better loss sequence on MNIST, SVHN and AGARICUS. It can be seen that the loss of DHPO is not dominant at the beginning, and then slowly drops to the lowest of all models. This corresponds to the two stages of DHPO in the actual training process. One is the hyper-parameter optimization stage, and the other is the training stage. DHPO is always looking for the best hyper-parameters in the beginning, which caused the loss dropping very slowly. Later, the optimal hyper-parameters are fixed, and the loss drops quickly then. However, the epochs are too small for WINE and CAR. As a result, the second stage has not been conducted enough yet. As for IRIS, its distribution is simple. So a small number of epochs is satisfied.

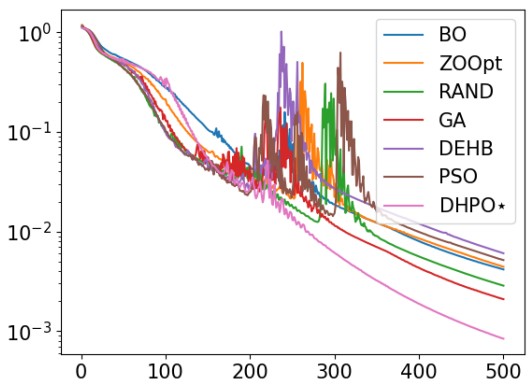

Figure 5: Loss sequence comparison in 500 epochs training.

Further, we set EPOCH to 500 and conduct HPO on WINE again. The results and the loss sequence are shown in Appendix E and Figure 5, respectively. It is easy to see from the figure that the DHPO takes an absolute advantage in the later stage of training, and finally achieves a good performance.

## 4.4 CASE STUDY

In this case study, we track DHPO for a whole optimization with running DNN on MNIST by recording hyper-parameters generated by DHPO after each epoch. And we observe whether hyper-parameters are really optimized in DHPO.

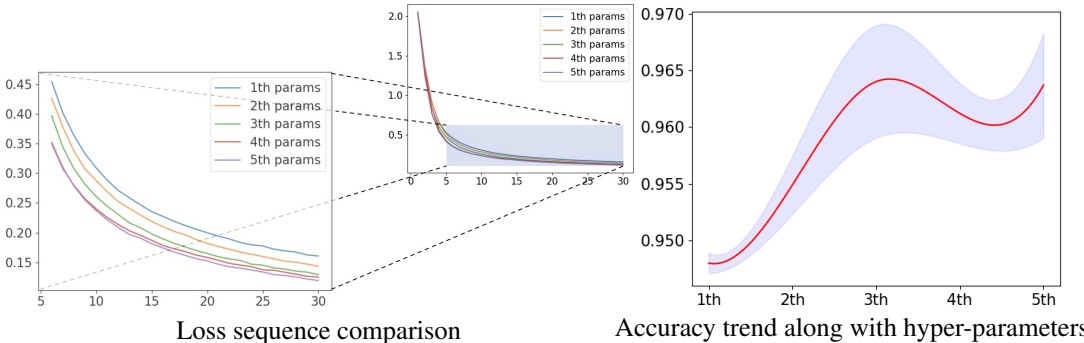

Loss sequence comparison          Accuracy trend along with hyper-parameters

Figure 6: case study on MNIST

In some singleton of DHPO, five groups of hyper-parameters are given out by DHPO along with the training session. We evaluate these five groups of hyper-parameters on DNN chronologically and visualize the performances in Figure 6. The later hyper-parameters have stronger convergence capabilities according to the loss. Performance(accuracy mean) and stability(accuracy std) under these hyper-parameters also shows an increasing trend. As the training session progresses, the quality of hyper-parameters gradually increases, which demonstrates that DHPO is an effective HPO method.

## 5 RELATED WORK

From a large scale, our work belongs to automatic machine learning. With the emergence of large-scale machine learning models, manual tuning (Montavon et al., 2012; gri, 2007) is no longer applicable. While random algorithm (ran, 2012) and grid search (Hinton, 2010) are ineffective with the target of performance and space, respectively. Goldberg uses evolutionary algorithm to optimize hyper-parameters. These algorithms have a good effect in exploring the distribution of the objective function value with the sample points and jumping out of the local optimal solution if we ignore the time consuming. A more stable optimization now belongs to zeroth-order optimization (Liu et al., 2018) and Bayesian optimization algorithm (Snoek et al., 2012), whose algorithms are complementary in principle. BO fits the distribution between samples and objective function with Gaussian process regression. This is suitable for the case where objective is differentiable. Zeroth-order optimization is more suitable for non-differentiable situations, and can solve the situation with multiple local optimal. However, these algorithms still need to repeatedly run the model to be adjusted, which brings a large time consuming. In order to solve this problem, the HyperBand (Li et al., 2016) changed the verification process. Epochs in a training session grow from 1 along with iterations. And a part of samples is eliminated after each iteration. Only one sample survives in the end. DEHB (Awad et al., 2021) combines the differential evolution and HyperBand for further optimization. On the whole, all current HPO algorithms rely on multiple training sessions to obtain feedback to adjust hyper-parameters. Only DHPO can optimize the hyper-parameters in the single training session, which is undoubtedly a meaningful work for automatic machine learning.

## 6 CONCLUSION & FUTURE WORK

In this paper, DHPO solves the time efficiency problem of HPO, and we also mentioned that DHPO is less affected by the initialization parameters. As long as a certain number of rounds, it can always overcome some more difficult hyper-parameters. Furthermore, we plan to study the why to apply DHPO on more hyper-parameters and more models rather than just neural network.

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

## A    META INFORMATION OF DATASETS

Table 3: Dataset meta information

| Dataset | #Train/ #Test | #Attributes | #Class |
|---|---|---|---|
| IRIS | 120/ 30 | 4 | 3 |
| WINE | 142/ 36 | 13 | 3 |
| CAR | 1,382/ 346 | 6 | 4 |
| AGARICUS | 6,499 / 1,625 | 116 | 2 |
| MNIST | 60,000 /10,000 | 1*28*28 | 10 |
| SVHN | 73,257 /26,032 | 3*32*32 | 10 |

## B    PROOF ABOUT DARTS'S EFFECTIVENESS WHEN DEALING WITH LOW DIMENSIONAL CANDIDATES

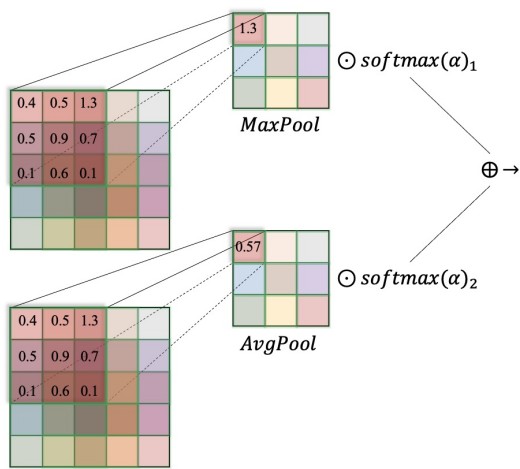

Figure 7: The forward propagation of pooling layer

We take Maximum pooling and average pooling as candidates. We will prove Darts is effective in this occasion. $\theta_{pool\_type} \in \{MaxPool, AvgPool\}$. $\boldsymbol{\alpha} \in \mathbb{R}^2$. According to Darts, the output of this pooling layer is calculated as formula 2 visualized in Figure 7.

$$\phi(X|\boldsymbol{\alpha}) = \frac{exp(\alpha_1)}{exp(\alpha_1) + exp(\alpha_2)} * MaxPool(X) + \frac{exp(\alpha_2)}{exp(\alpha_1) + exp(\alpha_2)} * AvgPool(X) \quad (2)$$

**Theorem 3.** *When $\alpha_i \gg \alpha_j, i \neq j$, $\boldsymbol{\alpha}$ is expressible for $\theta$ under $h(\boldsymbol{\alpha})$.*

$$h(\boldsymbol{\alpha}) = POOL\_CANDIDATES(\arg\max\{\boldsymbol{\alpha}\})$$
$$POOL\_CANDIDATES(0) = MaxPool \quad (3)$$
$$POOL\_CANDIDATES(1) = AvgPool$$

*Proof.* When $\alpha_1 \gg \alpha_2$, according to formula 3, $\theta_{pool\_type} = MaxPool$. $\mathcal{F}(X|\theta)$ represents a maximum pooling operation, namely $f(X|\theta) = MaxPool(X)$. Because $\frac{exp(\alpha_1)}{exp(\alpha_1) + exp(\alpha_2)} \approx 1$, $\frac{exp(\alpha_2)}{exp(\alpha_1) + exp(\alpha_2)} \approx 0$, then $f(X|\boldsymbol{\alpha}) \approx MaxPool(X)$. Assuming the proportion of error information is $\gamma$, $\gamma = \frac{exp(\alpha_2)}{exp(\alpha_1) + exp(\alpha_2)}$. With $\alpha_1 \gg \alpha_2, \gamma \to 0$. Then $f(X|\boldsymbol{\alpha}) \Leftrightarrow f(X|\theta)$ under $\gamma$. Proof is the same for $\alpha_2 \gg \alpha_1$. □

According to Theorem 1 and Theorem 3, as long as $\alpha_i \gg \alpha_j$, Darts is effective then. If $\gamma \leq 0.001$ is tolerant, then $|\alpha_1 - \alpha_2| > 6.9$ can meet it. It is easy when $\boldsymbol{\alpha} \in \mathbb{R}^2$. So Darts is effective in this occasion.

## C  EXPERIMENT RESULT ON SMALL DATASETS

Table 4: Time consuming and accuracy on small feature datasets.

| MODEL | TIME(s) | TOP1 | ALL MEAN | ALL STD | TOP5 MEAN | TOP5 STD | TOP10 MEAN | TOP10 STD |
|---|---|---|---|---|---|---|---|---|
| | | | | IRIS | | | | |
| BO | 45 | 100 | 99.67 | 0.58 | 100 | 0 | 100 | 0 |
| ZOOpt | 44 | 100 | 99.56 | 0.74 | 100 | 0 | 100 | 0 |
| RAND | 38 | 100 | 98.67 | 5.96 | 100 | 0 | 100 | 0 |
| GA | 148 | 98.89 | 99.81 | 0.44 | 100 | 0 | 100 | 0 |
| PSO | 142 | 100 | 94.76 | 15.88 | 100 | 0 | 100 | 0 |
| HB | 24 | 100 | 46.54 | 24.72 | 100 | 0 | 100 | 0 |
| DEHB | 48 | 100 | 99.33 | 2.21 | 100 | 0 | 100 | 0 |
| DHPO⋆ | 9 | 100 | 100 | 0 | 100 | 0 | 100 | 0 |
| DHPO† | 3 | 100 | | | | | | |
| | | | | WINE | | | | |
| BO | 46 | 99.07 | 94.38 | 5.05 | 98.33 | 0.37 | 98.06 | 0.5 |
| ZOOpt | 45 | 100 | 95.77 | 3.68 | 98.89 | 0.69 | 98.43 | 0.72 |
| RAND | 45 | 100 | 95.56 | 3.88 | 99.07 | 0.59 | 98.33 | 0.91 |
| GA | 151 | 96.3 | 96.45 | 1.62 | 99.26 | 0.37 | 98.89 | 0.56 |
| PSO | 145 | 99.07 | 77.18 | 21.38 | 97.78 | 0.74 | 97.5 | 0.59 |
| HB | 24 | 87.04 | 51.96 | 16.63 | 90.37 | 1.99 | 88.52 | 2.63 |
| DEHB | 45 | 100 | 94.1 | 7.03 | 98.52 | 0.74 | 97.87 | 0.83 |
| DHPO⋆ | 9 | 100 | 95.56 | 4.16 | 98.89 | 1.36 | 95.56 | 4.16 |
| DHPO† | 3 | 96.3 | | | | | | |
| | | | | CAR | | | | |
| BO | 52 | 93.26 | 91.02 | 4.98 | 92.99 | 0.15 | 92.86 | 0.17 |
| ZOOpt | 47 | 92.87 | 92.18 | 0.44 | 92.72 | 0.12 | 92.58 | 0.17 |
| RAND | 48 | 93.16 | 91.57 | 1.64 | 92.85 | 0.22 | 92.63 | 0.27 |
| GA | 171 | 93.55 | 92.45 | 2.37 | 93.47 | 0.19 | 93.38 | 0.16 |
| PSO | 171 | 93.35 | 90.06 | 8.77 | 93.43 | 0.11 | 93.33 | 0.13 |
| HB | 26 | 69.36 | 67.44 | 9.01 | 87.38 | 1.61 | 80.41 | 8.31 |
| DEHB | 49 | 93.26 | 90.89 | 5.82 | 93.12 | 0.12 | 93.02 | 0.14 |
| DHPO⋆ | 10 | 92.49 | 91.65 | 0.82 | 92.31 | 0.23 | 91.65 | 0.82 |
| DHPO† | 3 | 91.43 | | | | | | |
| | | | | AGARICUS | | | | |
| BO | 69 | 100 | 98.23 | 6.81 | 100 | 0 | 100 | 0 |
| ZOOpt | 71 | 100 | 100 | 0 | 100 | 0 | 100 | 0 |
| RAND | 64 | 100 | 99.99 | 0.03 | 100 | 0 | 100 | 0 |
| GA | 274 | 100 | 100 | 0 | 100 | 0 | 100 | 0 |
| PSO | 167 | 100 | 90.74 | 16.42 | 100 | 0 | 100 | 0 |
| HB | 40 | 99.94 | 58.13 | 15.74 | 99.94 | 0 | 99.93 | 0.01 |
| DEHB | 59 | 100 | 100 | 0 | 100 | 0 | 100 | 0 |
| DHPO⋆ | 16 | 100 | 100 | 0 | 100 | 0 | 100 | 0 |
| DHPO† | 5 | 100 | | | | | | |

## D  LOSS SEQUENCE ON IRIS, WINE AND CAR

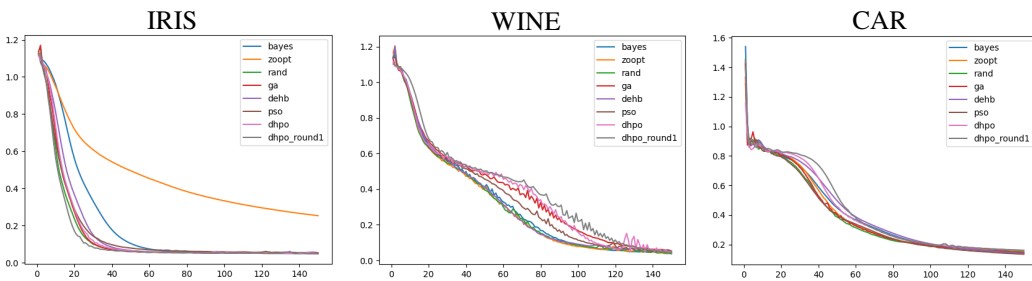

Figure 8: Loss sequence on three small datasets

## E  RESULT OF 500 EPOCH TRAINING ON WINE

|        | TIME | TOP1  | TOP5  | ALL   |
|--------|------|-------|-------|-------|
| BO     | 153  | 99.07 | 98.7  | 96.98 |
| ZOOpt  | 154  | 99.07 | 98.52 | 96.73 |
| RAND   | 151  | 99.07 | 98.52 | 96.98 |
| GA     | 556  | 99.07 | 99.63 | 97.27 |
| DEHB   | 160  | 100   | 98.89 | 97.31 |
| PSO    | 529  | 100   | 99.26 | 88.91 |
| DHPO⋆  | 36   | 100   | 100   | 98.06 |
| HB     | 24   | 87.04 | 90.37 | 51.96 |
| DHPO†  | 10   | 97.22 |       |       |

