# OpenReview forum: "Differentiable Hyper-parameter Optimization"
_ICLR.cc/2022/Conference — ICLR 2022 Submitted_

### Official Review · Reviewer_iUkA · 2021-10-28

**Correctness:** 3
**Technical Novelty And Significance:** 1
**Empirical Novelty And Significance:** 1
**Recommendation:** 1
**Confidence:** 5

**Main Review:**

Paper strengths:
+ The reparameterization of hyperparameters to make them derivable seems good. The idea is a good idea, even if already existing in the literature.

Paper weaknesses:
- This paper is not novel, a lot of existing work already tested this idea and showed that it already works. See the following references:

Franceschi, L., Donini, M., Frasconi, P. & Pontil, M.. (2017). Forward and Reverse Gradient-Based Hyperparameter Optimization. Proceedings of the 34th International Conference on Machine Learning, in PMLR 70:1165-1173

Stamoulis, D., Ding, R., Wang, D., Lymberopoulos, D., Priyantha, B., Liu, J., & Marculescu, D. (2019). Single-path nas: Designing hardware-efficient convnets in less than 4 hours. arXiv preprint arXiv:1904.02877.

Zhang, C., Ren, M., & Urtasun, R. (2018). Graph hypernetworks for neural architecture search. arXiv preprint arXiv:1810.05749.

Two of those references involve hypernetworks, with the same idea as proposed in this paper, which is define a larger network containing all subnetworks embedded within the hyperparameter search space.

- The selected benchmarks are outdated.
- Going back to the first point, the methods selected for comparison with the proposed approach are also outdated.
- The paper is poorly written, hard to read and the ideas are not clearly stated.


**Summary Of The Paper:**

The paper presents a gradient-based hyperparameter optimization method, wherein a derivable parameterization is proposed for various popular CNN hyperparameters including kernel size, number of channels and hidden layer size. Experiments show promise in terms of convergence speed when evaluated on simple benchmarks such as MNIST and SVNH, however this is irrelevant because a variant of this method already exists in the literature and nothing new is proposed.

**Summary Of The Review:**

This paper proposes ideas that are already existing in the literature, therefore it should be rejected.

---

### Official Review · Reviewer_Qr8L · 2021-11-01

**Correctness:** 3
**Technical Novelty And Significance:** 2
**Empirical Novelty And Significance:** 2
**Recommendation:** 3
**Confidence:** 4

**Main Review:**

Using a modified version of softmax to select channel size and kernel size seems novel which is not achieved by DARTS. However, there exist previous works on neural architecture search which can do this (e.g., Wan, Alvin, et al. "Fbnetv2: Differentiable neural architecture search for spatial and channel dimensions." Proceedings of the IEEE/CVF Conference on Computer Vision and Pattern Recognition. 2020.). Without comparison with these previous works, it is hard to appreciate the contribution and novelty of this proposed method.

Besides, the experiments are conducted only on MNIST and SVHN which are small scale by the standard in this topic. Considering that most papers in neural architecture search report performance on  ImageNet, it is important that the authors also report the performance there.

**Summary Of The Paper:**

This paper proposes to tune hyperparameters in a differentiable way by using a modified version of softmax function. It covers the tuning for three kinds of hyper-parameters as examples: channel size, kernel size and hidden layer size. Experiments on MNIST and SVHN (which are small scale datasets by modern DNN standards) show improvements on previous methods. The proposed method itself is reasonable, however, this paper misses many previous works on this topic, which makes it hard to appreciate its novelty.

**Summary Of The Review:**

In summary, the proposed method is ok but this paper ignores important previous works, which makes it hard to appreciate the contribution.

---

### Official Review · Reviewer_CR1f · 2021-11-02

**Correctness:** 1
**Technical Novelty And Significance:** 1
**Empirical Novelty And Significance:** 2
**Recommendation:** 3
**Confidence:** 5

**Main Review:**

Strengths:
1. On MNIST and SVHN, the proposed method has the best top1-error and the lowest STD. It also has the fastest convergence rate.

Weaknesses:
1. The authors claimed that they proposed the first model to solve hyper-parameter optimization in only one training session. However, they also said Darts could select the hyper-parameters in a single training session. Even if their method handled the argmax problem of DARTS, SNAS [1] has also considered this problem by the Gumbel-softmax trick. Hence, using differentiable methods is not their contribution.

2. The proposed theorems are trivial and meaningless. The authors try to handle the cases where multiple candidates occupy similar weights in DARTS. So, the theorem should prove that. At least, it is necessary to prove the three key characteristics in Section 3.1.3.

3. The paper didn't explain their method well. They just introduced their reparameterization. What is its differential? It is not a trivial problem [2]. Why it can solve the DARTS problem? It cannot be explained just in examples. Moreover, there are many writing issues and typos. For example,
(1) Formula 1 should be written in a sentence.
(2) The sentence "It is selecting an activation for the fully connected layer and supposing Relu is the best choice." is weird.
(3) What does the sentence "Firstly, we define found sufficient conditions as a new relation between \alpha and \theta." mean?

4. The experiments only used small data. The performance of the proposed method is not clear on the big data, like ImageNet. At least, CIFAR10 and CIFAR100 are needed. Further, this method should also work for other tasks not only for classification tasks.

5. The baselines are old and not enough. The baselines of Bayes optimization should include the first prize of Black Box Optimization competition in NeurIPS 2020, HEBO [3]. The SOTA of multi-fidelity methods should include BOHB [4] or BOSS [5].

[1] Xie, S., Zheng, H., Liu, C., & Lin, L. (2018). SNAS: stochastic neural architecture search. arXiv preprint arXiv:1812.09926.
[2] Franceschi, L., Donini, M., Frasconi, P., & Pontil, M. (2017, July). Forward and reverse gradient-based hyperparameter optimization. In International Conference on Machine Learning (pp. 1165-1173). PMLR.
[3] Cowen-Rivers, A. I., Lyu, W., Wang, Z., Tutunov, R., Jianye, H., Wang, J., & Ammar, H. B. (2020). Hebo: Heteroscedastic evolutionary bayesian optimisation. arXiv e-prints, arXiv-2012.
[4] Falkner, S., Klein, A., & Hutter, F. (2018, July). BOHB: Robust and efficient hyperparameter optimization at scale. In International Conference on Machine Learning (pp. 1437-1446). PMLR.
[5] Huang, Y., Li, Y., Ye, H., Li, Z., & Zhang, Z. (2020). An Asymptotically Optimal Multi-Armed Bandit Algorithm and Hyperparameter Optimization. arXiv preprint arXiv:2007.05670.



**Summary Of The Paper:**

The authors proposed a differentiable method to optimize neural networks and hyperparameters simultaneously like DARTS. And, they used a new parameterization to represent the original discrete hyperparameters. They claimed that it could eliminate the impact of rounding.

**Summary Of The Review:**

As mentioned in Main Review, the contribution of the paper is not enough. The novelty is not strong, since it is not the first model. The theorems are trivial and meaningless to describe the reparameterization, since the specific mapping is given. The experiments are not comprehensive enough considering the dataset and baselines. Moreover, the method itself is not explained well.

---

### Official Review · Reviewer_EFJq · 2021-11-03

**Correctness:** 2
**Technical Novelty And Significance:** 2
**Empirical Novelty And Significance:** 2
**Recommendation:** 5
**Confidence:** 3

**Main Review:**

Overall, this paper is a contribution in the right direction and has potential to cut down the hyperparameter optimization time greatly. The paper provides an elegant way to determine the optimum hyperparameters via a differentiable scheme.

Having said that, I have several concerns about the experiments.
1.	Another method which is closely related is Darts (Liu et al (2019)). Why it is not selected as a baseline in the comparisons?
2.	Two datasets MNIST and SVHN are used for the experiments. Both of them are similar and contain images of digits. It would be useful if the authors could show the performance of their method on other diverse datasets like CIFAR-10 or CIFAR-100.
3.	Section 4.2.3: the authors report the accuracy of optimized networks, but it is not clear if that is on the validation set or on the test set.
4.	Table 1: I don’t see much the benefit of the proposed method DHPO. Take the version DHPO+ as example. Compared to BLANK (i.e. train the network without hyper-parameter tunning), DHPO+ is just slightly better than BLANK, there is no significant difference. For example, 98.71% vs. 98.27% for MNIST and 85.05% vs 84.95% for SVHN.
5.	I am not sure why the authors use the version DHPO* (running 10 times) of their method when all the other methods are run only 3 times? Is this a fair comparison? If all of the methods including DHPO+ run the same 3 times, DHPO+ is not better than the baselines on datasets SVHN, Wine, and Car.
6.	Finally, it would be nice if the authors could apply their methods to neural architecture search using standard benchmarks e.g. NAS-101 etc. This will truly show the effectiveness of their method.

Minor Comments:
The paper seems to be written in a rush. The text and presentation could be significantly improved.
- the citation of Goldberg in the introduction section is not correctly done (missing publication year).
- space missing after BO and the following parenthesis, HB and the parenthesis, DHPO and the parenthesis.
-random search is cited incorrectly as (ran, 2012)!
- One page 2: incomplete sentence “While Darts can select the hyperparameters …”


**Summary Of The Paper:**


This paper introduces a method for hyper-parameter optimization (HPO) for deep neural networks. Its main idea is to replace hyper-parameters by trainable parameters, which can be included in the training process of the network itself.
The proposed method is applied to hyper-parameter tunning for two types of networks, namely CNN and FNN on two datasets MNIST and SVHN. It is also compared with several baselines.


**Summary Of The Review:**

To summarize, this paper addresses the HPO problem by transforming the hyper-parameters to trainable parameters and including them in the training process of the network. This approach can speed up the optimization time and allow the hyper-parameter optimization to happen in one single training session. Overall, the idea is interesting but the experiments are not convincing.

---

### Decision · Program_Chairs · 2022-01-20

**Decision:**

Reject

**Comment:**

The paper presents a gradient-based hyperparameter optimization method, wherein a differentiable reparameterization is proposed for various popular CNN hyperparameters including kernel size, number of channels and hidden layer size.

All reviewers have pointed out the lack of novelty (such reparameterizations are standard) and lack of convincing experiments.

The authors didn't write any rebuttal.

Overall, there is a large consensus among the reviewers that this paper is not ready for publication at ICLR.